# Gene Expression Analyses Reveal Mechanisms of Inhibited Yellowing by Applying Selenium-Chitosan on Fresh-Cut Broccoli

**DOI:** 10.3390/foods11193123

**Published:** 2022-10-08

**Authors:** Gang Ren, Yaping Liu, Bing Deng, Yu Wang, Wenyan Lin, Yulei Zhang, Jianbing Di, Jiali Yang

**Affiliations:** College of Food Science and Engineering, Shanxi Agricultural University, Jinzhong 030800, China

**Keywords:** selenium-chitosan, fresh-cut broccoli, storage, yellowing, chlorophyll, transcription factors, gene expression

## Abstract

The yellowing of green broccoli is a phenomenon that indicates a serious deterioration of freshness. The green broccoli has been more popular than the yellow one, with its higher nutritional value. Chitosan coating has been employed in vegetables for green-keeping, owing to its functions of regulating chlorophyll metabolism and antioxidant defense. Furthermore, selenium was commonly utilized in the pre-harvest of fruit and vegetables as an antioxidant and chlorophyll regulator. However, there have not yet been concerns about the effects of selenium-chitosan on vegetable yellowing. This study first investigated the impact of selenium-chitosan on the quality of fresh-cut broccoli yellowing during storage by analyzing the chromatic aberration and phytochromes. Additionally, then, the gene expression related to chlorophyll metabolism (*POR*, *CAO*, *HO*, *CHLI*, *NYC1*), carotenoid metabolism (*VDE*, *CCS*, *LCYE*, *ZEP*, *HYD*), and transcription factors (*NAC92*, *ZIPPER*, *bHLH66*, *APL*, *PIF4*) were analyzed using the RT-qPCR technique. Test results showed that treatment with selenium-chitosan can slow down the reduction in h° (Hue angle values) and reduce ethylene release rate and respiration intensity. Via the molecular approach, it was further identified that this treatment could inhibit chlorophyll degradation and carotenoid biosynthesis, accompanied by lower expression levels of heme oxygenase (*HO*), chlorophyllide A oxygenase (*CAO*), violaxanthin de-epoxidase (*VDE*), β-carotene 3-hydroxylase (*HYD*), *NAC92*, basic leucine zipper (*ZIPPER*), *bHLH66*, *PIF4* and *APL*, and higher expression levels of magnesium chelatase subunit I (*CHLI*) and lycopene ε-cyclase (*LCYE*) genes. This work can be used to explore the molecular mechanism of selenium-chitosan in inhibiting the yellowing of fresh-cut broccoli. This study will be of great economic importance in marketing and export by increasing the shelf life of fruits and vegetables.

## 1. Introduction

Broccoli (*Brassica oleracea* L. var. *Italica*) is a variety of kale species in the Brassicaceae family with green flower bulbs as the product, containing various bioactive substances, including vitamin C and vitamin E, as well as many glucosinolates [1]. Moreover, research shows that the degradation products of glucosinolates in broccoli can reduce the risk of cancer [2,3,4]. With its rich nutrition and easy cooking, fresh-cut broccoli has become one of the most popular and healthy types of food in western and eastern countries. However, as a processed product, fresh-cut broccoli is injured by mechanical processing, which accelerates broccoli yellowing. Research shows that 24-epibrassinolide can alleviate broccoli postharvest yellowing and improve its antioxidant capacity [5], and cytokinin can ameliorate the postharvest yellowing of broccoli [6]. In addition, several postharvest treatments, such as melatonin treatment [7], phytosulfokine α treatment [8], heat and UV-C treatment [9], have been used to maintain the quality of fresh-cut broccoli.

Selenium is one necessary component for metabolic pathways in the human body [10,11], which has significant physiological effects, such as antioxidant, detoxification and immunity-boosting [12]. For instance, selenium is an essential component of glutathione peroxidase. The thioredoxin reductase plays a vital role as an antioxidant and helps protect cells from oxidative stress [13,14]. Previous studies have shown that selenium can improve the storage quality of postharvest vegetables [15,16]. In particular, concerning the preservation of broccoli, one study pointed out that selenium (2 mg kg^−1^) application in broccoli inhibited respiratory intensity and ethylene release, positively maintaining quality and improving sensory quality [17]. On the other hand, selenium has also been widely applied in pre-harvest treatments [18,19,20]. Selenium can control chlorophyll and antioxidant systems and reduce salt-induced antioxidant pressure [18]. Ali et al. [20] addresses the recent advances in selenium arsenic to improve the growth and photosynthesis of melons under pressure. It also regulates antioxidant system and genetic expression of chlorophyll synthase (CHLG) and protochlorophyllide oxidoreductase (POR).

As a natural preservation material, chitosan is non-toxic and degradable, which has been used in the preservation of various fruit and vegetables, and its preservation effect is widely recognized [21,22,23]. For example, phenylalanine lyase (*PAL*) activity was significantly increased in fresh grapes treated with 1.0% chitosan. In addition to its direct effect on gray mold fungus, chitosan also produced other effects that helped in reducing decay [24]. Chitosan inhibited enzymatic browning and antioxidant enzyme activities [25], which delayed the ageing and energy depletion of pummelo fruit [26]. In addition, chitosan can regulate chlorophyll biosynthesis and antioxidant defense system in many plants, thus, allowing them to counter adverse stress responses [27,28].

Beneficial effects of selenium–chitosan complexes have been clearly demonstrated in animal studies. For example, adding selenium-chitosan to the diet increased mice’s blood selenium concentration and antioxidant capacity. Additionally, the immunity level of cells and body fluids in mice was significantly increased [29]. Meanwhile, broiler chickens fed a diet containing selenium-chitosan had lower thiobarbituric acid activity loss by cooking and higher muscle water retention than the control group, improving meat quality and blood parameters [30]. In addition, one study points out that chitosan-coated selenium nanoparticles for mRNA delivery have potential applications in tumor vaccines and immunotherapy [31], which was very beneficial for people’s health in anti-tumour treatments.

Although several studies have indicated the good food preservation properties of selenium-chitosan, little information has been reported regarding its impact on fresh-cut broccoli’s yellowing. Accordingly, this research aims to view how selenium-chitosan inhibits the yellowing of fresh-cut broccoli. The results of this study are expected to provide important information for the use of selenium-chitosan in the storage of fresh-cut broccoli and other fresh-cut vegetables.

## 2. Materials and Methods

### 2.1. Plant Material and Treatments

In this experiment, fresh broccoli without damage on its surface was selected as access material from the Juxin Entrepreneurship Park of Shanxi Agricultural University.

Before the formal experiment, a pre-experiment was performed. Through this pre-experiment, we determined the appropriate concentration of selenium (8 mg L^−1^) in chitosan selenium solutions.

Chitosan (1 g) was dissolved in 100 mL of 1% (*v*/*v*) glacial acetic acid solution at room temperature (26 °C) with magnetic stirring. Immediately after, *sodium selenite* (Na_2_SeO_3_) was added to the substrate to obtain a selenium-chitosan solution. Distilled water was used to wash the impurities on the surface of the buds. Then, the broccoli was dried and cut into 2 cm size buds. The samples were randomly divided into three groups. Fresh-cut broccoli without any treatment was used as the control group (CK) and two experimental groups were set up, including 1.0% chitosan (CS), 8 mg L^−1^ sodium selenite + 1.0% chitosan (Se+CS). After soaking for 10 min, the cut broccoli was removed and dried under a shade. The control and treatments group were placed in polyethylene bags (thickness 0.03 mm) and stored at 20 °C. Triplicates of each experiment were performed.

### 2.2. Chromatic Aberration

The color of broccoli was examined using a handheld colorimeter (YS3060, MOON-BJ, Shenzhen, Guangdong, China). The degree of hue (h°) was determined with the help of a colorimeter and used to estimate the color change during the freshness of fresh-cut broccoli. The colorimeter must be calibrated with black and white standard plates at the beginning of each measurement. The colorimeter automatically generates a* (redness and greenness) and b* (yellowness and blueness) values during the official measurement. The three fixed positions of broccoli balls were measured at each test and repeated three times. Hue degree (h°) was calculated as h° = tan^−1^ (b*/a*) when a* > 0 and b* > 0, or as h° = 180° − tan^−1^ (b*/a*) when a* < 0 and b* > 0 [32].

### 2.3. Ethylene Release Rate and Respiratory Intensity

The content of C_2_H_4_ and CO_2_ were measured in each respiration chamber every 24  h for 1  min with a gas analyzer (F-940, Felix, Camas, WA, USA). First, a certain amount of sample was weighed and breath samples were collected by placing a sample in a closed container for 5 h.

### 2.4. Determination of Chlorophyll Content

Chlorophyll was extracted and determined by Lichtenthaler and Wellburn’s method [33]. Exactly 1 g of sample was added with 7.5 mL of acetone and grounded thoroughly in a mortar. The mixture rested for 60 min at dark environment, followed by centrifugation at 8000× *g* for 15 min at 4 °C. The supernatant was then recovered and brought to a final volume of 25 mL with acetone. Absorbance values of solution were determined at wavelengths of 470, 645 and 663 nm, respectively. The following equations determined the total chlorophyll content.
Total chlorophyll (g × kg^−1^) = 20.2A_645_ + 8.02A_663_(1)

### 2.5. Determination of Carotenoids and Their Components

The treatment method was the same as described above (2.4). The content of carotenoids is determined by the following equation.
Chlorophyll a (g × kg^−1^) = 12.7A_663_ − 2.69A_645_(2)
Chlorophyll b (g × kg^−1^) = 22.9A_645_ − 4.68A_663_(3)
(4)$$\mathrm{Carotenoids}\;(g\;\times \;{\mathrm{kg}}^{-1})=\frac{1000\;(A_{470})-\;3.27\cdot \mathrm{chl}\;a\;-\;104\cdot \mathrm{chl}\;b}{229}$$

The extraction and fraction analysis of carotenoids were performed according to [34] with appropriate modifications. Approximately 0.5 g of sample was extracted by methanol/trichloromethane/distilled water (1:2:1). After 10 min centrifugation at 10,000× *g*, the chloroform phase was collected and the aqueous phases were re-extracted with chloroform 2 ± 3 times until residue was colorless. The combined chloroform phases were dried under a gentle nitrogen stream; the residue was dissolved in 6% (*w*/*v*) KOH in methanol and incubated in darkness environment at 60 °C for 30 min to complete saponification. The chloroform phase evaporates to dry in a stream of nitrogen.

The HPLC analyzed components of carotenoids with the C18 column, respectively (2 m × 2.1 mm × 55 mm). The chromatography was carried out at 25 °C. The mobile phase was methanol/ethyl acetate/acetonitrile (1:1:3, v:v:v). The flow rate was 15 µL s^−1^ and the injection volume was 10 µL. The standard curve method was used to quantify the fractions of carotenoids. Standard curves were generated using the standard phospholipids with lutein, β-carotene and carotenoid as an external standard.

### 2.6. Quantitative Real-Time PCR (RT-qPCR) Assay

Total RNA was isolated from broccoli by TransZol Up Plus RNA Kit (P11021, TransGen Biotech, Beijing, China) following the manufacturer’s protocol and quantified with a Nanodrop spectrophotometer (Nano Drop One/One^c^, GENE COMPANY LIMITED, Xianggang, China). Total RNA was reverse-transcribed to cDNA using a cDNA reverse transcription kit (R223, Vazyme, Nanjing, Jiangsu, China) following the manufacturer’s instructions. According to the ChamQ^TM^ SYBR^®^ Color qPCR Master Mix (Without ROX) kit (Q421-01/02/02, Vazyme, Nanjing, Jiangsu, China), the cDNA template was quantitatively analyzed by fluorescence quantitative PCR instrument to determine the relative expression level of the target genes. For RT-qPCR, the total volume was 20 μL, which included a cDNA template (1.2 μL), each primer (0.4 μL,10 mM), a 2× ChamQ™ SYBR^®^ Color qPCR Master Mix of 10 μL and RNase-Free water (8.0 μL). The thermal cycling conditions were 95 °C for 30 s, followed by 40 cycles at 95 °C for 10 s, and then 60 °C for 30 s. Melt curves (0.5 °C increments in a 55–95 °C range) for each gene were performed to assess the sample for non-specific targets, splice variants and primer dimers. Table 1 shows the primer designs for the genes to be tested [35]. The Actin2 (*ACT2*) gene was used as the internal reference. Each set of samples was replicated 3 times and the relative gene expression was calculated using the 2^−ΔΔCT^ method [36].

### 2.7. Statistical Analysis

Statistical and linear regression analysis were performed using Origin Pro 2021 (Origin Lab Inc., Northampton, MA, USA). SPSS v.20 software (SPSS Inc., Chicago, IL, USA) calculated standard deviation, one-way ANOVA, and Pearson correlation analysis.

## 3. Results

### 3.1. Chromatic Aberration Comparison between Treatments

It is well known that a* designates the color values on the red–green axis, b* designates the color values on the blue–yellow. In addition, higher values of hue angles represent more retention of green color or a diminished senescence in broccoli, and smaller values of hue angle represents a broccoli with more senescence or a minor retention of green color. The a* value of broccoli in the Se+CS group is lower than control, and the growth rate of a* was prolonged during storage. Notably, on the 5th day, the a* values of broccoli in the CS and Se+CS group remained negative by −1.06 and −1.2 (Figure 1A). The b* value of broccoli in the Se+CS group was less than control and only 62.4% of the control group. The h° value of fresh-cut broccoli in the Se+CS group was higher than those in the control group (Figure 1C). The results indicated that Se+CS treatment could better maintain fresh-cut broccoli’s appearance (green).

### 3.2. Analysis of Pigment Content

We further investigated the effect of Se+CS treatment on the pigments of fresh−cut broccoli. Figure 2 shows that the chlorophyll content of the Se+CS group was higher than that in the control group, while the carotenoid range of the Se+CS group was lower than that in the control group. Additionally, the content of lutein and the rate of β−carotene declines in the Se+CS group was lower than that in other treatment groups (Figure 2B,C). Compared with control and CS group, Se+CS treatment showed a positive effect on chlorophyll maintenance and reducing carotenoid synthesis.

### 3.3. Analysis of Ethylene Release Rate and Respiration Intensity

In Figure 3, we monitored the ethylene production and respiration intensity of control−, CS− and Se+CS−treated fresh−cut broccoli groups. There was no difference in ethylene release between the Se+CS group and the CS group during storage, but the ethylene release of the treatment group was lower than control group in later stage (Figure 3A). The respiration intensity showed marked diminish during the storage (Figure 3B), which may be due to the closed environment, where high CO_2_ concentration inhibited the respiration of broccoli. As a respiratory climacteric vegetable, Se+CS treatment delayed the emergence of respiratory peaks in broccoli (Figure 3B). The results showed that Se+CS treatment could inhibit fresh−cut broccoli’s respiration and ethylene production.

### 3.4. Analysis of Differentially Expressed Genes in Chlorophyll Metabolism

Previous studies reported that the expression of *POR*, *CHLI* and *HO* genes was continuously down−regulated during the broccoli yellowing [37], indicating that the expression of *POR* and *CHLI* genes was inhibited during broccoli yellowing. On the other hand, the expression level of *CAO* and *NYC1* genes was continuously up−regulated. RT−qPCR analyses revealed that the *POR* gene was induced on the 1st day after Se+CS treatment (Figure 4A). Meanwhile, the transcription levels of *CAO* in fresh−cut broccoli treated with Se+CS were inhibited during the yellowing process (Figure 4B). As shown in Figure 4C, the *HO* expression was low during mid-storage. In contrast, transcript levels of *CHLI* were increased by Se+CS treatment (Figure 4D). Se+CS treatment can slow the down−regulation of *NYC1* expression (Figure 4E).

### 3.5. Analysis of Differentially Expressed Genes in Carotenoid Synthesis

Luo et al. [38] revealed that *VDE*, *LCYE* and *ZEP* were down−regulated, and *CCS* and *HYD* were up−regulated during the yellowing process of fresh−cut broccoli. The results showed that the relative expression levels of *VDE*, *ZEP* and *HYD* after Se+CS treatment were lower than those in the control group and CS group (Figure 5A–C). Interestingly, although the expression levels of *LCYE* after Se+CS treatment were considerably higher than those in the control group, the gene of *LCYE* remains in a repressed state (Figure 5D). At the early stages of storage, Se+CS treatment had noticeable promoting effect on the *CCS* gene (Figure 5E).

### 3.6. Analysis of Transcription Factors in Pigment Metabolism

During the metabolism of pigments, the expression of related enzyme genes is also affected by many regulatory factors. Previous studies have shown that these regulatory factors are positively correlated with the face of *BoCAO* and *BoHYD* [38]. Having demonstrated that, we investigated the transcription factors. Our data indicate that the Se+CS treatment exhibited an inhibitory effect on *NAC92*, *ZIPPER*, *bHLH66*, *APL* and *PIF4* regulatory factors, and the overall inhibitory effect was better than the other fractions (Figure 6). However, there were no detectable differences on the expression of *NAC92*, *APL* and *bHLH66* between Se+CS treatment and CS treatment (Figure 6A–C). In summary, selenium treatment inhibited the expression of specific essential enzyme genes by suppressing the expression of transcription factors, thus, suppressing the yellowing of fresh−cut broccoli.

## 4. Discussion

Currently, studies on selenium−chitosan have focused on effects on people and animals, but there is a little information of this treatments on plants. Available analysis for the combination of selenium (0.02 ppm) and chitosan (1.5%) treatment of guava effectively protected the integrity of cell membranes [39]. Selenium controls gray mold and reduces disease [40]. Based on the results of existing studies, this study was conducted to investigate the effect of selenium−chitosan on the yellowing of fresh−cut broccoli. The results of this study show that broccoli treated with selenium−chitosan is greener than that treated with single−chitosan treatment. Through assessment of a* and b* values, we found that the yellowing of fresh−cut broccoli was irreversible during storage, but the use of selenium delayed the yellowing process. Furthermore, the color of broccoli may be due to chlorophyll, carotenoids and other significant substances [38]. Broccoli yellowing results from the continuous degradation of chlorophyll and a gradual increase in carotenoids, eventually leading to yellowing. The chlorophyll degradation and carotenoid biosynthesis of broccoli in the Se+CS group were the slowest among all experimental groups. Ethylene has been reported to accelerate the breakdown of chlorophyll [41]. Moreover, Lu et al. [42] also indicated that ethylene is required to degrade chlorophyll in citrus peels, demonstrating that yellowing increases the tissue content of ethylene. Based on the above studies, the ethylene content of broccoli in different treatments was further investigated. The result shows that the Se+CS group was lower than the control group. However, there was a subtle difference between the Se+CS and CS groups, which indicated that chitosan film for barrier properties of ethylene would not be altered by adding selenium. Additionally, zeaxanthin, lutein, β−carotene, and antheraxanthin were prevalent carotenoids in white corn [43]. Two types of lycopene cyclase regulate lutein and β−carotene. Here, we find that the rate of lutein synthesis and β−carotene breakdown is slower, suggesting that secondary metabolites in carotenoid synthesis during broccoli yellowing are affected by Se+CS treatment.

Chlorophylls are the primary plant pigments affecting leaf color formation in higher plants [44]. Interestingly, broccoli undergoes yellowing accompanied by chlorophyll synthesis and degradation process [37]. On the one hand, *CHLI*, *POR* and *CAO* are three critical enzymes in the chlorophyll synthesis sub−pathway [45]. In addition, the *HO* gene is an essential enzyme in the biliary production pathway. This study showed that the chlorophyll content first increased and then decreased, indicating that the chlorophyll metabolism flux downstream of *the protoporphyrin IX* node is tilted to the chlorophyll synthesis during the storage, and then gradually tilted toward the heme degradation sub−pathway. RT−qPCR results revealed that expression of *CLHI* was promoted by Se+CS treatment, indicating that Se+CS treatment could promote chlorophyll biosynthesis. Yang et al. [46] indicated that *POR* was the essential functional protein in regulating chlorophyll metabolism. In addition, *POR* catalyzes proto−chlorophyllide to form the chlorophyllide, and the different kinds of *POR* iso−forms were found in *Arabidopsis*, barley and cucumber [47,48]. The experiment results supported that the *POR* expression of fresh−cut broccoli was increased during storage by Se+CS treatment, suggesting that Se+CS treatment could retard chlorophyll degradation by up−regulating *POR* (Figure 4A). This effect, however, has only been clearly detected on first day. *NYC1* plays a vital role in chlorophyll degradation [49], and chlorophyll b reductase was encoded by *NYC1* that catalyzes the degradation of chlorophyll b to 7−hydroxymethyl chlorophyll a [50]. *CAO* is a gene involved in the chlorophyll biosynthesis pathway and is up−regulated during color change [51]. Interestingly, the expression of *CAO* was inhibited after Se+CS treatment in the later stage, thus, increasing content of Chl a (Figure 4B).

Carotenoids are responsible for fruit and vegetables’ yellow, orange, and red color [52]. *LCYE* was a closely correlated lycopene cyclase protein that catalyzes the formation of ε−rings in the β, and ε−branch of the carotenoid biosynthesis pathway [53]. Similarly, Ren et al. [54] reported that the synthesis of lutein and β−carotene was controlled by the *LCYB* and *LCYE* genes. Still, only *LCYE,* as a differentially expressed gene, affects the content of lutein and β−carotene during broccoli yellowing [38]. From the detection of lutein and β−carotene content, the results showed that the content of lutein is higher than the content of β−carotene, which is consistent with the expression effect of the *LCYE* gene. It means that in the treatment of Se+CS, the carotenoid complex of broccoli prefers the α−carotene subgroup to the β−carotene subgroup. However, although most of the metabolic flow is on an α−carotene branch with increased lutein, the increasing rate of lutein is decelerated due to the suppression of the *HYD* gene expression.

In addition, overexpression of the *HYD* gene in rice led to a significant decrease in β−carotene content and an increase in violaxanthin and other lutein contents [55]. Moreover, as noted by Sestili et al. [56], simultaneous inhibition of *LCTE* and *HYD*, as an efficient method, can increase the β−carotene content in wheat while this research reveals a repressive effect of Se+CS treatment on the expression of *HYD* (Figure 5E). The genes of *VDE* and *ZEP* are critical in the downstream pathways and are responsible for regulating the cycle between zeaxanthin and violet xanthin [57,58]. Luo et al. [38] reported that *VDE* was up−regulated, leading to the accumulation of zeaxanthin during broccoli yellowing. Our experimental results showed that *VDE* expression was lower in Se+CS treatment during storage, as expected (Figure 5A). The low expression of *HYD* might have led to a decrease in zeaxanthin, which promoted the conversion of violaxanthin to zeaxanthin. However, lower expression of VDE inhibited the phenomenon.

Meanwhile, Se+CS treatment also inhibited the expression of *ZEP*, which slowed down the transformation of zeaxanthin to violaxanthin (Figure 5D). This resulted in a dynamic balance between zeaxanthin and violet xanthin. Moreover, the functionally related carotenoid cyclase enzyme is *CCS*, which catalyzes the final step in the carotenoid pathway and converts the violaxanthin into neoxanthin [59]. Furthermore, the *CCS* is also a key enzyme for capsaicin synthesis in broccoli. Our results showed that Se+CS treatment had a motivating effect on the *CCS* gene before turning yellow, and the effect was not apparent after turning yellow (Figure 5B).

The yellowing of fruit and vegetables is also influenced by several transcription factors. Transcriptional sequence analysis revealed 46 transcription factors involved in chlorophyll metabolism, chloroplast development and division, photosynthesis and other pigment biosynthesis [50]. The *bHLH* family, one of the most influential families of transcription factors, plays an essential role in plant growth, vegetative growth and reproductive growth [60]. Furthermore, multiple *bHLH* genes include *PIF4*, *bHLH57*, *bHLH66*, *bHLH74* and *bHLH75*, which participate in flowering time regulation, hormonal regulation and chloroplast development [61]. At the same time, *NAC2* and *EIN3a* could activate the transcription of genes related to carotenoid biosynthesis during the ripening of papaya fruit [13]. A qualitative study by Luo et al. [38] described that *BobHLH66*, *BoPIF4*, *BoZIPPER*, *BoNAC92* and *BoAPL* encode transcription factors that are potential regulators of the broccoli pigment metabolism pathway. The results showed that the use of CS treatment had an inhibitory effect on these regulatory factors. After comparative analysis, the inhibitory effect was enhanced by adding selenium to chitosan (Figure 6). This also confirmed that selenium−chitosan treatment could effectively inhibit the expression of *NAC92*, *ZIPPER*, *bHLH66* and *APL* transcription factors in broccoli, thereby achieving inhibited of expression of *CAO* and *HYD*. Similarly, the expression of *PIF4* was also inhibited by treating with Se+CS. Additionally, the study by [62] pointed out that *PIF4*, as a transcriptional activator, up−regulates the expression of *NYE1* to trigger the degradation of chlorophyll and inhibits the chlorophyll biosynthesis and chloroplast activity by repressing the transcription of *PORC* and *GLK2*.

## 5. Conclusions

The most prominent finding to emerge from this study was that selenium−chitosan treatment effectively maintained the green phenotype of fresh−cut broccoli. The treatment retarded chlorophyll degradation and carotenoid synthesis. In addition, we also investigated the mechanism of Se+CS treatment in delaying the yellowing of fresh−cut broccoli. The results showed that Se+CS treatment could affect chlorophyll degradation by down−regulating *CAO* and *HO*, and up−regulating *CHLI* and *NYC1*. Meanwhile, Se+CS treatment also could inhibit carotenoid synthesis by down−regulating the expression of *VDE* and *HYD* and up−regulating the expression of *LCYE*. We found that Se+CS treatment could inhibit the expression of transcription factors (*NAC92*, *ZIPPER*, *bHLH66*, *APL*, *PIF4*), which related to *HYD* and *CAO*, and finally achieve the purpose of yellowing inhibition.

## Figures and Tables

**Figure 1 foods-11-03123-f001:**
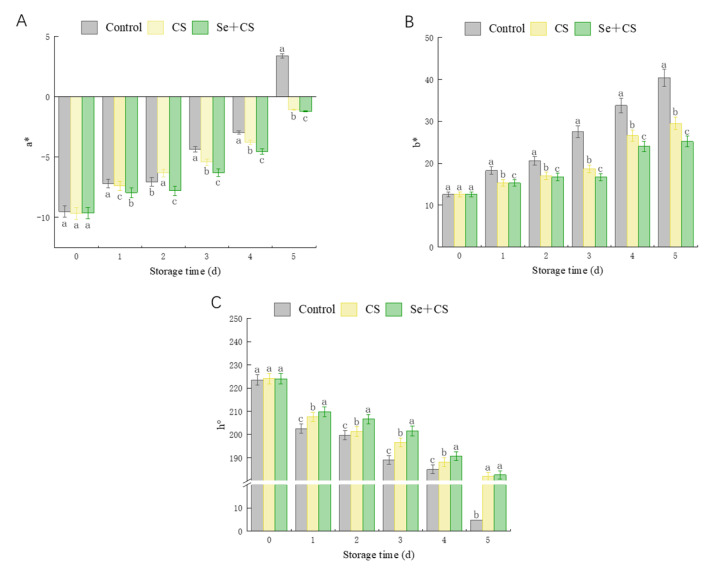
Hunter’s color value in fresh−cut broccoli treated with Se+CS during storage. (**A**). a*. (**B**). b*. (**C**). h°. Different letters represent significant differences in the same group during storage (*p* < 0.05) by Duncan’s multiple range test.

**Figure 2 foods-11-03123-f002:**
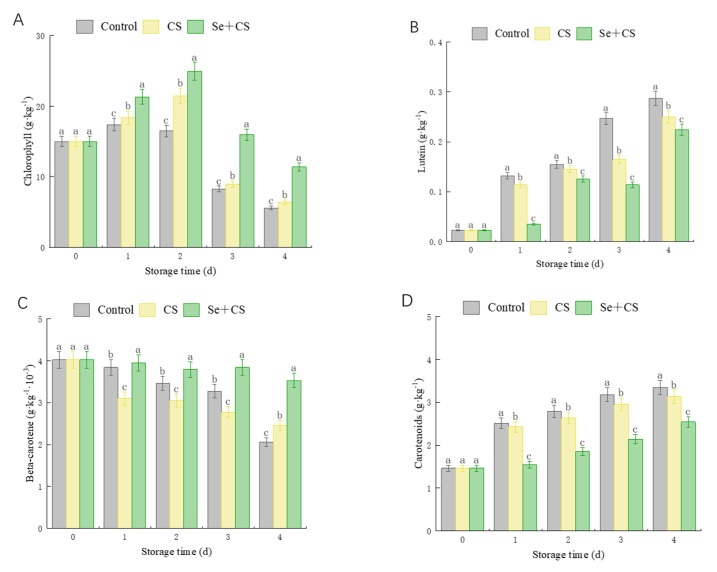
Pigment content for broccoli treated with Se+CS during storage (**A**). Chlorophyll content. (**B**). Lutein content. (**C**). β−carotene content. (**D**). Carotenoid content. Different letters represent significant differences in the same group during storage (*p* < 0.05) by Duncan’s multiple range test.

**Figure 3 foods-11-03123-f003:**
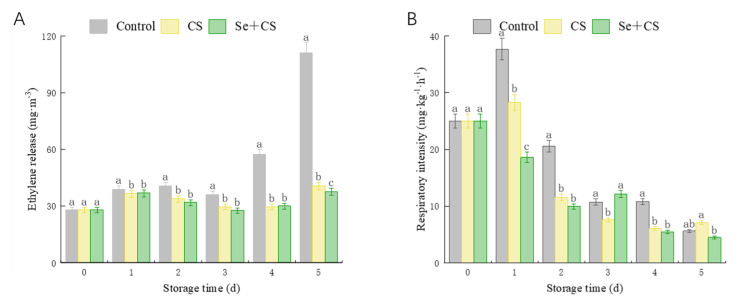
Ethylene production and respiration for broccoli treated with Se+CS during storage. (**A**). Ethylene production. (**B**). Respiratory intensity. Different letters represent significant differences in the same group during storage (*p* < 0.05) by Duncan’s multiple range test.

**Figure 4 foods-11-03123-f004:**
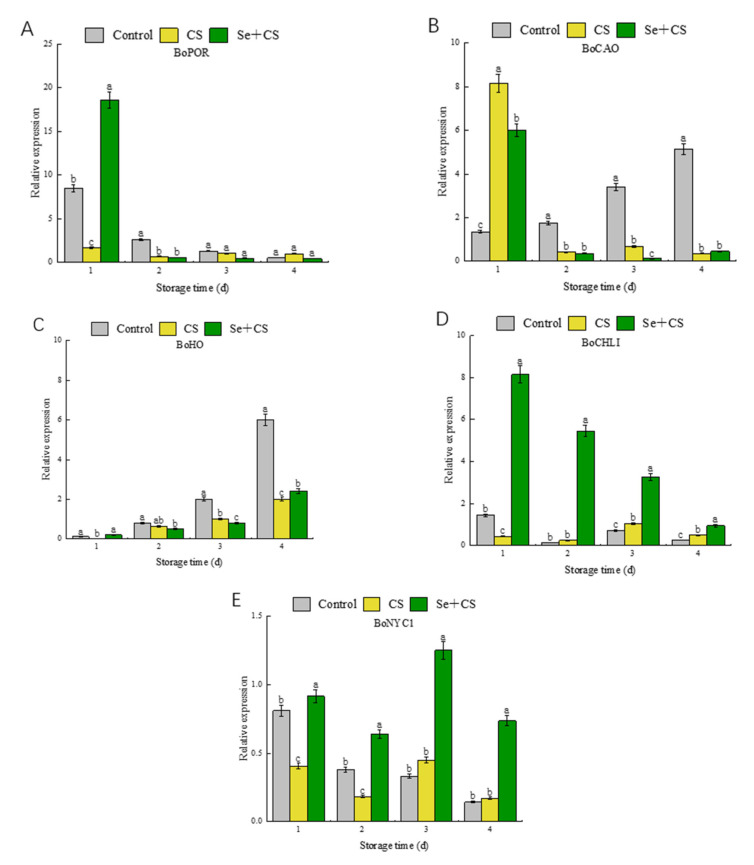
Relative gene expressions of chlorophyll metabolism-related genes in broccoli treated with Se+CS during storage. (**A**). *BoPOR*. (**B**). *BoCAO*. (**C**). *BoHO*. (**D**). *BoCHLI*. (**E**). *BoNYC1*. Different letters represent significant differences in the same group during storage (*p* < 0.05) by Duncan’s multiple range test.

**Figure 5 foods-11-03123-f005:**
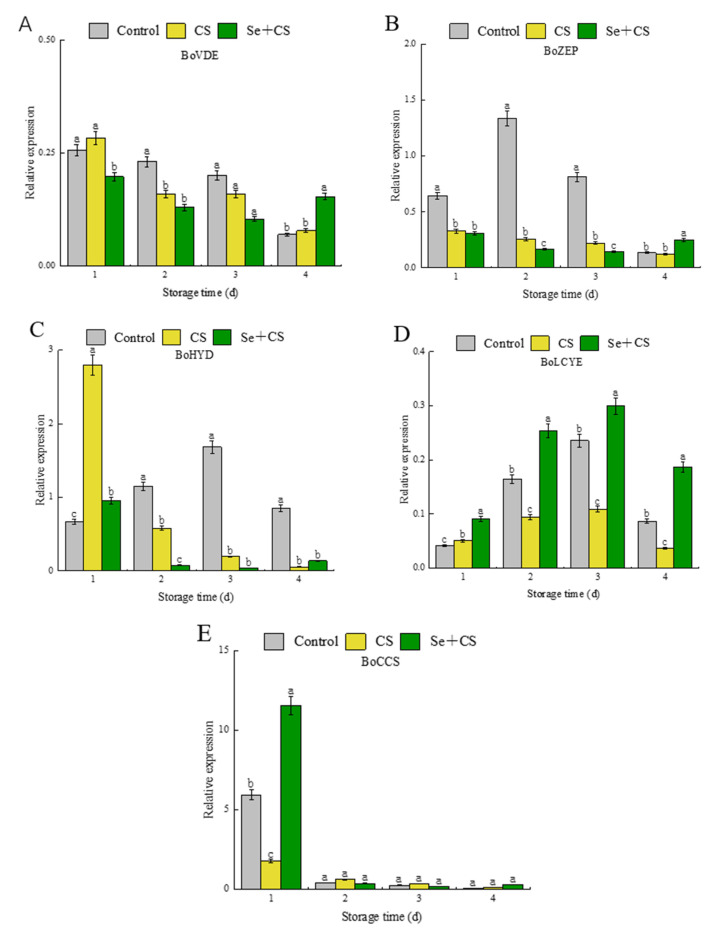
Relative gene expressions of carotenoid synthesis-related genes in broccoli treated with Se+CS during storage. (**A**). *BoVDE*. (**B**). *BoCCS*. (**C**). *BoLYCE*. (**D**). *BoZEP*. (**E**). *BoHYD*. Different letters represent significant differences in the same group during storage (*p* < 0.05) by Duncan’s multiple range test.

**Figure 6 foods-11-03123-f006:**
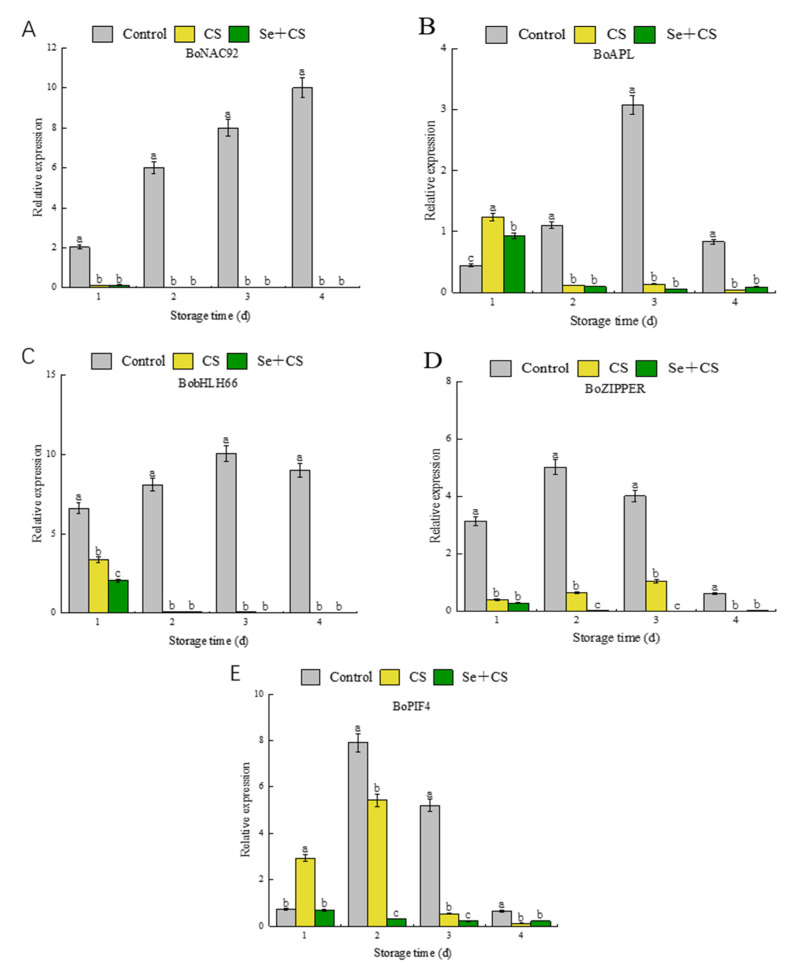
Relative gene expressions of pigment metabolism-related transcription factors in broccoli treated with Se+CS during storage. (**A**). *BoNAC92*. (**B**). *BoZIPPER*. (**C**). *BobHLH66*. (**D**). *BoAPL*. (**E**). *BoPIF4*. Different letters represent significant differences in the same group during storage (*p* < 0.05) by Duncan’s multiple range test.

**Table 1 foods-11-03123-t001:** Name and sequence of primers designed for genes related to broccoli yellowing.

Gene	Gene Function	Primer Sequences (5′-3′)	Gene Bank Accession
*ACT2*	Internal reference	F: 5′-TTCAAGCTGGAGCCAAGAAGGTTC-3′ R: 5′-ACGAATGGTGCGAGACAGTTAGTG-3′	AF044573
*BoHO*	Involvement in chlorophyll metabolism	F: 5′-CCACAACTCACCAGCTTCCACTTC-3′ R: 5′-GCCGTCGTAGCTGCAATCACC-3′	LOC106295105
*BoCAO*		F: 5′-ACCATCATGCTCCTTCACGACAAG-3′ R: 5′-TCGCCAACTCTCCGCCTCTG-3′	LOC106350863
*BoNYC1*		F: 5′-ACATTGTGACGATGACGAGCACTG-3′ R: 5′-TAACTCCACCGAGCCAGCTATACC-3′	LOC106341846
*BoCHLI*		F: 5′-CGGCGAGACTGACGAAGTGAAC-3′ R: 5′-AGCAAGCGGTATCGAGAGCAATC-3′	LOC106305850
*BoPOR*		F: 5′-TTCAGGCTGCTTACTCGCTTCTTC-3′ R: 5′-CAACCATGGCATTGGCGTGAAG-3′	LOC106307221
*BoHYD*	Involvement in carotenoid synthesis	F: 5′-AGAGGCTTCTCGGTCTGCTACG-3′ R: 5′-GCTCGACATCACTGCGGCTATTAG-3′	LOC106302033
*BoZEP*		F: 5′-AGACGGCGGCGGAGAGTAAG-3′ R: 5′-GCTCAAGTCCTTCTCGAACACCAG-3′	LOC106390346
*BoCCS*		F: 5′-CTGGCTATCGCTGATCCTTGGC-3′ R: 5′-GTTCTCCGCCGTACTTCTCATCG-3′	LOC106296637
*BoLCYE*		F: 5′-AAGGCAGCGAAAAGCAGGAA-3′ R: 5′-TGACCATCCATGTAGTTTCTCCG-3′	LOC106296030
*BoVDE*		F: 5′-GATACGGCGGTGCGGTTGTG-3′ R: 5′-CTCTCCACCAGAGGAGGTTCAGG-3′	LOC106415200
*BoAPL*	Transcription factors for pigment metabolism	F: 5′-GCAGCCGCACAAGGAGTATGG-3′ R: 5′-AGCTGTTCGTGCAACCTTCTCTG-3′	LOC106402384
*BoNAC92*		F: 5′-AACGACAAGACCTCAAGCACATCC-3′ R: 5′-ACACTCACAAGAGAACGCTCCAAC-3′	LOC106294382
*BoPIF4*		F: 5′-GTGATGACCGTTGGACCGAACC-3′ R: 5′-ACCAGAGGAGCCACCTGATGATG-3′	LOC106328266
*BobHLH66*		F: 5′-CCGCCTCCGTCCTCAGATGG-3′ R: 5′-ATTCATCATTCCACCTGCCGTTCC-3′	LOC106308180
*BoZIPPER*		F: 5′-CTACCACGTCCGATGAAGCAACTG-3′ R: 5′-GAAGCCAAGCAACCTGCGAGAG-3′	LOC106325179

## Data Availability

The data presented are contained within the article.

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
