# Peer review of "Gene Expression Analyses Reveal Mechanisms of Inhibited Yellowing by Applying Selenium-Chitosan on Fresh-Cut Broccoli"

_foods, 2022, doi:10.3390/foods11193123_

Round 1
Reviewer 1 Report
The current manuscript is a good work dealing with “RT-qPCR gene expression analyses reveal mechanisms of inhibited yellowing by applying selenium-chitosan on fresh-cut broccoli”
The authors are responsible for the following comments.
o In order to improve the m.s, I encourage the authors, deeply correct the English edition.
- For the title, it will be better to remove (RT-qPCR) to be "Gene expression analyses reveal mechanisms of inhibited yellowing by applying selenium-chitosan on fresh-cut broccoli"
o In the abstract, in line 19, insert the name of genes after its category "chlorophyll metabolism (POR, CAO, HO, CHL1, NYC1) and carotenoid metabolism (insert the name of genes), as well as transcription factors (mention the name of genes)………
o At the end of the abstract it will be better if the authors explored the economic importance of the used in this study in marketing and export by increasing the shelf life.
o In key words, change factor/factors and add gene expression
o In the introduction, in line 63 change helped reduce/ helped in reducing.
o From line 68 to line 76, rephrase the paragraph to be more clear and informative, in line 68, in human and animals what? What do you want to say, there is missing information.
o In M&M, in line 98, change the control group and treatment group/ the control and treatments group…..
o In M&M, 2.4., section from line 116 to line 122, rephrase.
o In M&M, 2.6., part, line 145, total RNA was isolated from broccoli fruits……
o In line 152, gene/ genes
o In line 160, cite the required reference.
o Refine the title of table 1 to be more informative
o In table 1, mention the accession number in gene bank of the used genes including reference gene
o Redraw table 1 to classify the studied gene to three groups, chlorophyll metabolism, carotenoid synthesis and transcription factors for pigment metabolism, beside the reference gene.
o In results, line 185, control group. While, the carotenoid …….
o In line 195, broccoli groups.
o In line 206 to line 216, please rewrite this part, why do you start this section with results from previous data represented in figure 4, it is not clear where the authors cited results in fig 4E before that in fig4D and the same in fig 4C.
o The previous comment is the same for 3.5., (from line 221 to line 229) and 3.6., (line 233 to line 243) of results section.
o Put the gene names in italic.
o Conclusion needs to be re-written keeping in view the actual findings of this study.
Reviewer 2 Report
In spite of the results being well presented, it would be relevant to display the existing visual differences between broccoli to explain the impact of chitosan use.
Comments
1.The problem of yellowing in broccoli as a clear sign of deterioration, which this research intends to fight against using selenium-chitosan.
2.It is relevant as this allows to increase the shelf-life of broccoli, as well as maintaining its nutritional value for a longer period. It is also interesting on giving a new usage to selenium-chitosan that is already used in other plant species.
3.It is not completely original as selenium-chitosan is already known and previously used in other plant species and similar purposes, as well as it approaches the extent of shelf life which is a thematic that has a lot of research undergoing. On the other hand, there are no studies using selenium-chitosan against yellowing in broccoli so there is some originality.
3.The use of the RT-qPCR gene expression provides a more robust source of data comparing with the other published materials on plant species that tend to focus more on biochemical analytics.
4.The paper is well written. And, the text is clear and easy to read.
5.The conclusions are consistent with the evidence and arguments presented. In fact, the conclusions sound more like the discussion than conclusions. I would like to refer that probably a feature of the following steps on the research would improve the conclusions.
6.The authors address the main question posed.
Round 2
